

# Effects of low oxygen concentrations on aerobic methane oxidation in seasonally hypoxic coastal waters

Lea Steinle[1,2], Johanna Maltby[2,3], Tina Treude[2,4], Annette Kock[2], Hermann W. Bange[2], Nadine Engbersen[1], Jakob Zopfi[1], Moritz F. Lehmann[1], Helge Niemann[1,5]

[1]Department of Environmental Sciences, University of Basel, 4056 Basel, Switzerland
[2]GEOMAR Helmholtz Centre for Ocean Research Kiel, Marine Biogeochemistry Research Division, 24148 Kiel, Germany
[3]Department of Natural Sciences, Saint Joseph's College, Standish, Maine, USA
[4]Department of Earth, Planetary & Space Sciences and Atmospheric & Oceanic Sciences, University of Los Angeles, Los Angeles, California, USA
[5]CAGE – Centre for Arctic Gas Hydrate, Environment and Climate, Department of Geology, UiT the Arctic University of Norway, 9037 Tromsø, Norway

*Correspondence to*: Lea Steinle (lea.steinle@unibas.ch)

**Abstract.** Coastal seas may account for more than 75% of global oceanic methane emissions. There, methane is mainly produced microbially in anoxic sediments from where it can escape to the overlying water column. Aerobic methane oxidation (MOx) in the water column acts as a biological filter reducing the amount of methane that eventually evades to the atmosphere. The efficiency of the MOx filter is potentially controlled by the availability of dissolved methane and oxygen, as well as temperature, salinity, and hydrographic dynamics, and all of these factors undergo strong temporal fluctuations in coastal ecosystems. In order to elucidate the key environmental controls, specifically the effect of oxygen availability, on MOx in a seasonally stratified and hypoxic coastal marine setting, we conducted a 2-year time-series study with measurements of MOx and physico-chemical water column parameters in a coastal inlet in the southwestern Baltic Sea (Eckernförde Bay). We found that MOx rates always increased toward the seafloor, but were not directly linked to methane concentrations. MOx exhibited a strong seasonal variability, with maximum rates (up to 11.6 nmol $l^{-1}$ $d^{-1}$) during summer stratification when oxygen concentrations were lowest and bottom-water temperatures were highest. Under these conditions, 70–95% of the sediment-released methane was oxidized, whereas only 40–60% were consumed during the mixed and oxygenated periods. Laboratory experiments with manipulated oxygen concentrations in the range of 0.2–220 μmol $l^{-1}$ revealed a sub-micromolar oxygen-optimum for MOx at the study site. In contrast, the fraction of methane-carbon incorporation into the bacterial biomass (compared to the total amount of oxidised methane) was up to 38-fold higher at saturated oxygen concentrations, suggesting a different partitioning of catabolic and anabolic processes under oxygen-replete and oxygen-starved conditions, respectively. Our results underscore the importance of MOx in mitigating methane emission from coastal waters and indicate an organism-level adaptation of the water column methanotrophs to hypoxic conditions.



## 1 Introduction

Methane is a potent greenhouse gas, but the contributions of individual natural sources to the atmospheric budget are still not well constrained (Kirschke et al., 2013). Coastal (shelf) seas are estimated to account for more than 75% of the global marine methane emissions, even though they cover only about 15% of the total ocean surface area (Bange et al., 1994; Bakker et al.,

2014). In coastal systems, methane is mainly produced via degradation of organic matter by methanogenic archaea in anoxic sediments (Bakker et al., 2014). Part of the produced methane is consumed via anaerobic- or aerobic oxidation of methane within the sediments (Knittel and Boetius, 2009; Boetius and Wenzhöfer, 2013), but a significant fraction often escapes into the overlying water column (Reeburgh, 2007). In the water column, methane can also be oxidized anaerobically in the rare case of water column anoxia, but most of it is consumed via aerobic oxidation of methane (MOx; R1), mediated by aerobic

methane-oxidizing bacteria (MOB; Reeburgh, 2007).

$$CH_4 + 2O_2 \rightarrow CO_2 + 2H_2O \qquad\qquad (R1)$$

MOx is hence an important sink for methane before its potential release into the atmosphere, but little is known about the efficiency of MOx in shallow coastal ecosystems, where the distance between the sediment source and the atmosphere is short, leaving little time for exhaustive methane oxidation during vertical advective or turbulent-diffusive transport. Coastal

ecosystems undergo large temporal variations with regard to e.g., temperature, oxygen, salinity, or organic matter input (e.g., Lennartz et al., 2014, Gelesh et al., 2016). Additionally, over the past decades, the number of seasonally or permanently hypoxic or even anoxic coastal zones has increased worldwide, most often as a consequence of anthropogenic eutrophication and/or climate change (Diaz and Rosenberg, 2008; HELCOM: 2009; Rhein et al., 2013; Lennartz et al., 2014; Rabalais et al., 2014). Model results also predict that such oxygen-depleted zones will expand in the future since increasing surface water

temperatures will lead to enhanced stratification and, thus, to less oxygen supply to bottom waters (Keeling et al., 2010; Friedrich et al., 2014). Several definitions for water column oxygenation levels were suggested in the literature (Diaz and Rosenberg, 2008; Canfield and Thamdrup, 2009; Middelburg and Levin, 2009; Naqvi et al., 2010). Here, we adopt the threshold adapted by Middelburg and Levin (2009) and Naqvi et al. (2010), where hypoxia is defined as $[O_2] < 63\ \mu mol\ l^{-1}$.

Multiple environmental factors can affect MOx in seasonally stratified coastal marine environments. Oxygen concentrations,

for instance, are likely to impact MOx, which is an aerobic or in some cases micro-aerobic process (Carini et al., 2005; Schubert et al., 2006; Blees et al., 2014). Hence, the organic matter flux and, directly related, the rate of oxygen consumption due to organic matter remineralisation may influence how much methane is oxidized via MOx. Enhanced organic matter input can also increase methane production rates (Maltby 2015), which in turn may stimulate MOx rates. Moreover, increasing ocean water temperatures as a result of climate change may directly impact metabolic rates of microbes, e.g., of

methanogens or methanotrophs (Madigan et al., 2015). Increasing surface water temperature could also have an indirect effect on MOx, as it will influence water column stratification and thus oxygenation levels during summer time (Diaz and Rosenberg, 2008; Keeling et al., 2010; Friedrich et al, 2014). Finally, water mass circulation, transporting methanotrophs to- or away from methane sources can also affect MOx rates (Steinle et al., 2015). In order to predict the fate of coastal methane,





knowledge on the seasonality of water column methane oxidation, and the environmental controls thereupon, are of great importance for predicting future changes in methane emissions (Bakker et al., 2014).

The seasonally hypoxic Boknis Eck time series station located in Eckernförde Bay (SW-Baltic Sea) is an excellent site to investigate coastal water column MOx under different oxidation/stratification regimes (see e.g., Bange et al., 2010). Here,

we present results from a 2 years study, during which we investigated MOx rates, methane concentrations, and physicochemical parameters in the water column at Boknis Eck on a quarterly basis. The aim was to assess seasonal dynamics of, and the environmental controls on, MOx in this coastal inlet. Combining field observations and laboratory experiments, we specifically addressed the role of oxygen as modulator of MOx by determining minimum oxygen requirements and potential oxygen concentration optima.

## 2 Materials and methods

### 2.1 Site description

The time-series station Boknis Eck (54°31.15 N, 10°02.18 E; www.bokniseck.de; Fig. 1) is situated at the entry of the coastal inlet Eckernförde Bay in the SW-Baltic Sea, with an approximate water depth of 28m at the sampling site (Bange et al., 2011). Physico-chemical water column parameters have been measured regularly since 1957, making this station one of

the longest-operated marine time-series stations worldwide (Lennartz et al., 2014). The hydrography of Eckernförde Bay is characterised by the outflow of low-salinity Baltic Sea water and the inflow of higher-salinity North Sea water through the Kattegat and the Great Belt (Fig. 1a). From mid-March to mid-September, the water column is stratified with a pycnocline at ~15m water depth (Lennartz et al., 2014). Large phytoplankton blooms generally occur in early spring (Feb./Mar.) and in autumn (Sept.–Nov.), and minor blooms occasionally during summer (July/Aug.; Smetacek et al., 1984; Smetacek 1985;

Bange et al., 2010). The resulting high organic matter fluxes and respiration rates result in a high oxygen demand and, as a consequence, in bottom-water hypoxia (or occasionally even anoxia) during summer (Hansen et al., 1999). The frequency of water column hypoxia in Eckernförde Bay has increased since the 1960s (Lennartz et al, 2014). The high organic matter sinking flux and rapid sedimentation of organic matter also lead to enhanced methanogenesis in the muddy sediment, and to high methane concentrations in the overlying water column (Jackson et al., 1998; Whiticar 2002; Treude et al., 2005; Bange

et al., 2010). Results from monthly samplings during the last ten years have revealed year-round methane seepage from the seafloor and methane super-saturation (with respect to the atmospheric equilibrium) of surface waters (Bange et al., 2010).

The stratification period in Eckernförde Bay ends in fall (Oct/Nov), with the onset of surface-water cooling and water-column mixing during fall storms (Bange et al., 2010). Besides these seasonal water-column stability changes, episodic perturbations of the water column can be observed during occasional major saltwater injections from the North Sea. Such

events occur over short time periods (days to weeks) when easterly winds are followed by strong westerly winds, leading to the inflow of salty, oxygen-enriched (fall/winter) or oxygen-poor (summer) bottom water, respectively (Nausch et al., 2014;





Mohrholz et al., 2015). As a result of variable exchange (not only during major salt-water injections) between North Sea- and Baltic Sea water, bottom water salinity varies strongly between 17 and 24 psu (Lennartz et al, 2014).

## 2.2 Sampling

Sampling was conducted every three months over a time-period of two years (Oct. 2012–Sept. 2014). On board RV *Alkor*,
RC *Littorina* or RB *Polarfuchs*, water-column samples were collected from 1, 5, 10, 20, and 25 meters below sea level (mbsl) using a rosette sampler equipped with $6 \times 4$ L Niskin bottles and CTD and $O_2$ probes for continuous measurements of conductivity, temperature, density, and dissolved oxygen, respectively (Hydro-Bios, Kiel, Germany; $O_2$-sensor: RINKO III). In the following we use the common term CTD for the combined suite of sensors, including the $O_2$ sensor. We note that water column zones where the CTD oxygen sensor indicated 0 $\mu$mol $l^{-1}$ $O_2$ are not necessarily anoxic in the strict sense.
Given that the detection limit of the Winkler method applied to calibrate the CTD oxygen sensor is 1-2 $\mu$mol $l^{-1}$, micro-aerobic concentrations at the sub-$\mu$M levels were likely not detected as such. Water aliquots were sampled for measurements of methane concentration and MOx activity. For an overview of sampling dates and corresponding parameters or incubations see Table 1. In the following, samples from 1 and 5 mbsl will be referred to as "surface waters", and samples from 20 and 25 mbsl are considered "bottom waters".

## 2.3 Dissolved methane concentration

For dissolved methane (hereafter just methane) determinations, three 25 ml vials per sampling depth were filled bubble-free immediately after CTD-rosette recovery, poisoned with saturated mercury chloride solution (50 $\mu$l) and stored at room temperature. Methane concentrations were then determined by gas chromatography and flame ionization detection with a headspace method as described in Bange et al. (2010).

## 2.4 Methane oxidation rate measurements

MOx rates ($r_{MOx}$) were determined in quadruplicates from ex situ incubations of water samples with trace amounts of $^3$H-labeled methane as described in (Steinle et al., 2015) based on a previously described method (Reeburgh et al., 1991). In brief, 25 ml crimp-top vials were filled bubble-free and closed with bromobutyl stoppers that are known to not affect MOx activity (Helvoet Pharma, Belgium; Niemann et al., 2015). Within a few hours after sampling, 6 $\mu$l of a gaseous $^3$H-CH$_4$/N$_2$
mixture (~15 kBq, <30 pmol CH$_4$, American Radiolabeled Chemicals, USA) were added and samples were incubated for 3 days in the dark at in situ temperature. The linearity of MOx during a time period up to 5 days was verified in selected samples by replicate incubations at 24, 48, 72, 96, and 120 hours. At the end of the incubation, we determined the $^3$H activities of the H$_2$O ($A_{H_2O}$; includes possible radio-label incorporation into biomass) and of non-reacted CH$_4$ ($A_{CH_4}$) by wet scintillation. Activities were corrected for (insubstantial) tracer turnover in killed controls (addition of 100 $\mu$l saturated
HgCl$_2$ solution). From these activities, we calculated the fractional turnover of methane (first order rate constants; $k$; Eq. 1):





$$k = \frac{A_{H_2O}}{(A_{H_2O} + A_{CH_4})} \times \frac{1}{t} \qquad (1)$$

where $t$ is incubation time.

$r_{MOx}$ was then calculated from $k$ and water column [$CH_4$] at the beginning of the incubation (see Sect. 2.4; Eq. 2):

$$r_{MO_x} = k \times \left[CH_4\right] \qquad (2)$$

$r_{MOx}$ from incubations with $^{14}$C-labelled methane (see Sect. 2.4.1) was determined analogously by measuring the radioactivity of $^{14}CH_4$, $^{14}CO_2$ ($A_{CO_2}$), and the remaining radioactivity ($A_R$,), according to Blees et al. (2014) and Steinle et al. (in press). The use of $^{14}$C also allows the assessment of methane incorporation into biomass (Eq. 3):

$$C_{incorp.} = \frac{A_R}{A_{CO_2} + A_R} \qquad (3)$$

**2.4.1 Oxygen manipulation experiments**

Oxygen dependency experiments were conducted with water from 20 mbsl sampled in Feb. 2014 and Jun. 2014. Until the start of the experiments (~5 days after sampling), the sampled water was stored headspace-free at 0 °C. For the incubations, the water was filled into 160 ml glass vials, which were closed with gas-tight bromobutyl rubber stoppers (Helvoet Pharma, Niemann et al., 2015). Before use, the stoppers were boiled in water and stored under $N_2$ in order to avoid bleeding of oxygen from the rubber matrix into the vials. The vials were then purged for 30 min with high-purity $N_2$ gas, transferred to

an anoxic glovebox ($N_2$ atmosphere), and the $N_2$ headspace that was generated during the purging was exchanged with $N_2$-purged Boknis Eck water. For final oxygen concentrations <15 µmol l$^{-1}$ (Fig. 3), we injected (in the glovebox) 0.1–8 ml of Boknis Eck water that was previously equilibrated with ambient air. For final oxygen concentrations >15 µmol l$^{-1}$, 10 ml headspace with a predefined gas mixture of $O_2$:$N_2$ (5:95–12:88) was added, and the vials were left to equilibrate overnight at 4 °C. Additional vials were used to fill up the headspace the next day. After the oxygen adjustment, we added 12 µl of

anoxic Boknis Eck water enriched in methane resulting in a final methane concentration of ~100 nmol l$^{-1}$ in the incubation vial. For $r_{MOx}$ determination, we added $^3$H-$CH_4$ tracer in the glove box as described above. Oxygen concentration was measured with a high-sensitivity (detection limit ca. 0.005 µM) OX-500 microsensor (Unisense). $N_2$-purged seawater amended with dithionite (final concentration 9 mmol l$^{-1}$) was used as blank. Oxygen concentrations during incubations were determined by measuring initial and final oxygen concentration in parallel incubations, which were amended with 6 µl $N_2$

gas instead of $^3$H-$CH_4$. Vials were incubated at 20 °C in the dark for 6 h ([$O_2$] >15 µmol l$^{-1}$) or 10 h (for [$O_2$] <15 µmol l$^{-1}$). Incubation times differed because of time constraints during sample processing. $r_{MOx}$ was determined as described above.



### 2.4.2 Temperature dependency experiments

Similar to the determination of $r_{MOx}$ at in situ temperature, we incubated water column aliquots in triplicates at different temperatures between 0.5 and 37 °C in incubators. We determined the temperature dependency of MOx in samples collected at 5 and 20 mbsl between Sept. 2013 and Sept. 2014 (Table 1). The MOx temperature optimum is defined here as the

temperature where measured MOx rates were highest (i.e., not as the optimum for growth).

### 2.5 Areal MOx rates and methane release to the atmosphere

For each sampling date, we interpolated linearly between the measured rates at different depths, to obtain depth-integrated average rates for the 28m deep water column, and to calculate the methane oxidation rate per unit area ($F_{MOx}$: in µmol m$^{-2}$ d$^{-1}$). Methane flux ($F_{atm}$; in µmol m$^{-2}$ d$^{-1}$) from surface waters to the atmosphere was calculated according to Bange et al.

10   (2010):

$$F_{atm} = k_w \times 60 \times 60 \times 24 \times (Sc_{CH_4}/600)^{-0.5} \times ([CH_4] - [CH_4]_{eq}) \qquad (4)$$

where $Sc_{CH_4}$ is the Schmidt number, which is dependent on temperature and salinity, and is calculated as the ratio of kinematic viscosity of seawater (Siedler and Peters, 1986) and the diffusion coefficient of methane in seawater (Jähne et al., 1987). $[CH_4]$ is the measured methane concentration at 1 mbsl, $[CH_4]_{eq}$ is the calculated $CH_4$ equilibrium concentration

(Wiesenburg and Guinasso, 1979), with respect to average atmospheric pressure at sea-level (measured at the lighthouse Kiel) and the atmospheric methane concentration (median between 2012 and 2014) in the Northern Hemisphere (Mace Head Station; Prinn et al., 2014), and $k_w$ is the gas transfer coefficient. Following Bange et al. (2010), we used $0.65 \times 10^{-5}$ and $1.51 \times 10^{-5}$ m s$^{-1}$ as minimum and maximum values for $k_w$, respectively, as recommended for coastal systems (Raymond and Cole, 2001). These $k_w$ values already include wind strength.

To determine the strength of water column stratification, we calculated the Brunt–Väisälä buoyancy frequency N:

$$N = \sqrt{-\frac{g}{\rho} \times \frac{d\rho}{dz}} \qquad (5)$$

where g is the gravitational constant, $\rho$ is the potential density and z is the geometric height. N (in cycles per hour; cph) was calculated for the depth interval 10–20 mbsl.

### 3 Results

### 3.1 Seasonal variations of physico-chemical water column properties

Physico-chemical water column properties generally showed large seasonal variations (Fig. 2). Salinity varied strongly in bottom- (18–25 psu) and surface waters (13–24 psu) with highest salinities observed in Sept.–Nov. (Fig. 2a). During winter





samplings, water temperatures were coldest and rather invariant throughout the water column at 1–3 °C (Mar. 2013, Feb. 2014; Fig. 2c). They increased from spring until the end of summer to a maximum of 18°C in the surface (June 2013, Sept. 2014) and 13 °C in bottom waters (Oct. 2012, Sept. 2014). Water density at Boknis Eck showed the same seasonal pattern as salinity (data not shown). Bottom water densities varied between 1015 and 1019 kg m$^{-3}$ with the highest densities from June–

Nov. Surface water densities ranged between 1009 and 1015 kg m$^{-3}$, with lowest densities in June and Sept. Dissolved oxygen concentrations in bottom waters varied from <1 to 450 µmol l$^{-1}$, with the highest concentrations observed during fully mixed conditions (Mar. 2013/Feb. 2014). Bottom waters became hypoxic in June (2013/2014), and reached oxygen concentrations ≤ 1 µmol l$^{-1}$ towards the end of the stratification period (Sept. 2013/2014). In surface waters, levels of dissolved oxygen were always high (>280 µmol l$^{-1}$), reaching maximum concentrations in Mar. 2013/Feb. 2014 (>450 µmol

l$^{-1}$; Fig. 2d), when according to comparatively high chlorophyll a concentrations (data not shown) sampling took place during a phytoplankton bloom. Dissolved methane concentrations reached values up to 466 ± 13 nmol l$^{-1}$ (Mar. 2014) in bottom waters, and were always >30 nmol l$^{-1}$ during the time of our study (Fig. 2f). We did not observe a clear repeating seasonal pattern in the two years of our study. Surface dissolved methane concentrations ranged between 8 and 27 nmol l$^{-1}$, corresponding to super-saturation levels of 270–870 % with respect to atmospheric equilibrium (2.8–4.3 nmol l$^{-1}$).

**3.2 Seasonal variability of MOx**

First order rate constants ($k$) of MOx were always highest in bottom waters (0.003–0.084 d$^{-1}$) with the exception of Nov. 2013 when the highest $k$ was measured at 15 mbsl (Fig. 2g). The same pattern was found for $r_{MOx}$, which was always highest in bottom waters (1.0–11.6 nmol l$^{-1}$ d$^{-1}$), again with the exception of Nov. 2013 (Fig. 2h). The highest values for $r_{MOx}$ were observed in summer (June 2013) or fall (Oct. 2012/Sept. 2013/Sept. 2014), when bottom water temperatures were highest

and oxygen concentration were lowest. A correspondence between maximum $r_{MOx}$ and high/maximum CH$_4$ concentrations (i.e., in Mar. 2014) was not observed.

**3.3 Influence of oxygen concentrations on MOx activity**

During our lab-based experiments with adjusted oxygen concentrations, $r_{MOx}$ was always highest in incubations with the lowest oxygen concentration (Fig. 3). Rates under nearly saturated oxygen conditions were significantly lower (50% in

February, 75% in June; p <0.05) than the rate measured at the lowest initial oxygen concentration. However, while we found that $r_{MOx}$ did not vary strongly at different oxygen concentrations <140 µmol l$^{-1}$ (Fig. 3a) in Feb. 2014, the rates measured in June 2014 were more variable (Fig. 3b).

All incubations remained oxic during the incubation time, even incubations with very low initial oxygen concentrations (Fig. 3c, d). In Feb. 2014, 41–79% of oxygen were consumed during incubations with initial [O$_2$] <15 µmol l$^{-1}$ (Fig. 3c), and 14–

40% during incubations with initial [O$_2$] >15 µmol l$^{-1}$. In June 2014, 22–73% of the oxygen was consumed during incubations with initial [O$_2$] <15 µmol l$^{-1}$ (Fig. 3d), and 3–8% during incubations with [O$_2$] >15 µmol l$^{-1}$.




Similar to incubations with $^3$H-CH$_4$, $k$ values determined with $^{14}$C-CH$_4$ at saturated oxygen concentration were lower than $k$ values measured in incubations with [O$_2$] <0.5 µmol l$^{-1}$ (32% lower on average; Fig. 4a, Fig. S1a, b). In contrast, a higher fraction of the oxidized $^{14}$C was incorporated into biomass in incubations at saturated oxygen concentration than at [O$_2$] <0.5 µmol l$^{-1}$ (between 108 and 3800% more; Fig. 4b, Fig. S1c, d), irrespective of water depth and sampling season (Fig. S1).

Radio-label incorporation at low oxygen-levels ([O$_2$] <0.5 µmol l$^{-1}$) was generally more pronounced in bottom- (20 mbsl) than in surface waters (5 mbsl).

In the subsequent discussion of the experimental data, for the sake of simplicity, hypoxic conditions (i.e., [O$_2$] <63 µmol l$^{-1}$) will be referred to as "low" and [O$_2$] >63 µmol l$^{-1}$ as "high" oxygen concentrations.

### 3.4 Temperature dependence of MOx

In general, $k$ increased with temperature and reached maximum values at 20–37 °C, indicating a mesophilic temperature optimum (Fig. 5a,c; shown are results from Sept. 2013 and Feb. 2014; Table 1; Fig. S2). Only in Nov. 2013, maximum MOx was observed at 10–20 °C, consistent with a psychrophilic temperature optimum (Fig. 5b). These patterns were independent of water depth (Fig. 5, Fig. S2).

### 3.5 Water-column methane removal by MOx and methane fluxes to the atmosphere

Depth-integrated $r_{MOx}$ (= $F_{MOx}$) varied between 11.7 µmol m$^{-2}$ d$^{-1}$ (Mar. 2013) and 82.3 µmol m$^{-2}$ d$^{-1}$ (Sept. 2014; Table 2). Estimated average fluxes of methane to the atmosphere were 3.6–9.2 µmol m$^{-2}$ d$^{-1}$ during stratified periods, and 10.0–25.1 µmol m$^{-2}$ d$^{-1}$ during mixed periods (considering a minimum or maximum $k_w$, respectively; Table 2; Raymond and Cole 2001). The water column was only weakly stratified in Oct. 2012, Mar. 2013, and Feb. 2014 (i.e., N <120 cph), and strongly stratified during all other samplings (i.e., N >120 cph; Table 2).

## 4 Discussion

### 4.1 Seasonal variations at Boknis Eck

### 4.1.1 Development of seasonal hypoxia

Oxygen concentrations were always close to saturation levels during our winter samplings (i.e., Mar. 2013, Feb. 2014), when the water column was poorly stratified and phytoplankton blooms occurred, which is typical for this time period (Bange et

al., 2010). During June samplings, we observed much lower bottom water oxygen concentrations, indicating the onset of hypoxia, reaching sub-micromolar oxygen concentrations below 24 mbsl in September (2013, 2014). Our observation is in accordance with a previous time series study (2006–2008) by Bange et al. (2010), who found hypoxic events starting between May and August and lasting until September or November. Long-term monitoring at Boknis Eck showed that the frequency and length of hypoxic events have increased over the last twenty years (Lennartz et al., 2014), although nutrient





inputs into the Baltic Sea were strongly reduced (HELCOM, 2009). One of the main reasons for the on-going decrease in oxygen concentration is the increasing water temperature since the 1960s (Lennartz et al., 2014). Higher surface water temperatures have led to an extension of the stratification period (starting earlier in the annual cycle), reducing the overall exchange between bottom and surface waters (Hoppe et al., 2013; Lennartz et al., 2014). Furthermore, a general increase in

temperature also enhances mineralisation of organic matter in bottom waters and results in a higher biological oxygen demand (Hoppe et al., 2013; Lennartz et al., 2014).

### 4.1.2 Seasonal dynamics of methane concentrations and MOx

Methane concentrations at Boknis Eck were in a similar range as in other shelf seas and coastal ecosystems (e.g., Rehder et al., 1998; Bange, 2006; Upstill–Goddard 2016). We did not observe any clear seasonal methane concentration patterns as

observed for oxygen concentrations and other physico-chemical parameters (Fig. 2). Our data are consistent with observations from 2006–2008 by Bange et al. (2010), who showed that methane concentrations did not follow bimodal seasonal variations. Instead, increases in water column methane followed chlorophyll *a* concentrations in surface waters with a 1-month time lag, suggesting that pulses of elevated organic matter input to the sediments were boosting benthic methanogenesis.

We did not observe direct links between water column oxygenation and methane concentrations (Pearson linear correlation coefficient: $R^2 = 0.007$; two-tail Student's t-test: $p = 0.6$). However, water column oxygen concentrations can have indirect and inverse effects on methane concentrations: Low oxygen concentrations combined with high sedimentation rates of organic matter lead to enhanced burial of "fresh" organic matter to the sediments, which in turn favours methanogenesis at Boknis Eck (Bange et al., 2010; Maltby 2015). Furthermore, water column stratification at times of water column hypoxia at

Boknis Eck hinders exchange between bottom and surface waters, which further promotes methane accumulation in bottom waters. Such a modulating effect of oxygen- on methane concentrations has also been found in other hypoxic ecosystems (see review by Naqvi et al., 2010). Nonetheless, highest methane concentrations were not always observed during periods of minimum oxygen concentrations. On the contrary, we rather found reduced net methane accumulation in bottom waters at times of hypoxia and strong water column stratification (summer/autumn 2013, Fig. 2), likely caused by the comparatively

high MOx rates observed during this time period.

MOx rates at Boknis Eck were of the same order of magnitude as those measured in other coastal environments (e.g., Abril and Iversen 2002; Mau et al., 2013; Osudar et al., 2015). We did not observe a direct stimulus of high methane concentrations on MOx, that is, $k$ (and $r_{MOx}$) was not correlated with methane concentrations (Pearson linear correlation coefficient: $R^2 = 0.1$; two-tail Student's t-test: $p = 0.6$). For example, we found low methane concentrations in Sept. 2013 co-

occurring with high MOx rates, while high methane concentrations in Feb. 2014 were accompanied by only moderately elevated MOx rates. Previous studies suggested an inverse relationship between turnover time (i.e., $1/k$) and methane concentrations (Elliott et al., 2011; Nauw et al., 2015; Osudar et al., 2015; James et al., 2016). Although this relationship





may be robust across different environmental settings, several studies found that on a smaller, local scale the $CH_4$/MOx connection does not necessarily apply (Heintz et al., 2012; Mau et al., 2012; Steinle et al., 2015, in press).

MOx measurements in this study revealed seasonal dynamics, with lower rates during the winter season when the water column was mixed and oxygen concentrations were high (Mar. 2013, Feb. 2014), and highest rates during stratified, hypoxic

conditions (June–Oct.). Indeed, $k$ mainly correlated negatively with oxygen concentration (Pearson linear correlation coefficient: $R^2 = 0.45$; two-tail Student's t-test: $p = 0.001$), whereas this was not the case with temperature and methane concentration. In order to further investigate these putative links observed during the time-series study, we conducted laboratory experiments to specifically assess the effects of oxygen concentration (Sect. 4.2) and temperature (Sect. 4.3) on MOx.

**4.2 Aerobic methanotrophy under micro-oxic conditions**

We observed the highest MOx rates in Sept. 2013 and Sept. 2014 (Fig. 2f), when bottom water oxygen concentrations were below the detection limit of the oxygen sensor (i.e., 1–2 µmol l$^{-1}$). Whether oxygen concentrations reached true zero levels is unclear. We did not measure hydrogen sulphide concentrations yet the absence of any hydrogen sulphide smell is pointing to the fact that traces of oxygen were probably still present. Anaerobic oxidation of methane (AOM) thus seems unlikely to

account for the observed oxidation rates since anaerobic methanotrophs cannot persist in the presence of oxygen even at sub-micromolar levels (Treude et al., 2005; Knittel and Boetius, 2009). Other recent studies in lakes reported on the occurrence of aerobic methanotrophy in apparently anoxic environments (Blees et al., 2014; Milucka et al., 2015; Oswald et al., 2015). Two of these studies showed that, in the presence of light, photosynthetic algae create oxic microniches and provide enough oxygen for MOx to proceed under otherwise anoxic conditions (Milucka et al., 2015; Oswald et al., 2015). We cannot rule

out that in situ MOx in oxygen-depleted waters of Boknis Eck was, at least to some degree, fuelled by photosynthesis in bottom waters. However, light intensities deeper than 20 mbsl in the murky waters at Boknis Eck are likely too low to account for the observed high rates. Furthermore, in the dark-incubation experiments of this study, rates were still highest at the lowest oxygen concentration (Fig. 3), arguing against any significant photosynthetic production of oxygen to support light-dependent MOx. In another lake study, a large potentially active MOx community was discovered in the anoxic

hypolimnion ~200m below the lake surface, where light-dependent MOx can be excluded (Blees et al., 2014). It was hypothesized that sporadic oxygen-inflow was sufficient to sustain a viable MOx community in anoxic waters, well below the permanent redoxcline. At Boknis Eck, it is possible that episodic oxygen inputs through horizontal advection can occur, as has been observed after North Sea water inflows into the anoxic basins of a Danish fjord (Zopfi et al., 2001) and the Baltic Sea (Schmale et al., 2016). Although we cannot say without doubt what the sources of oxygen are that sustain MOx under

micro-oxic conditions our incubation experiments with oxygen concentrations as low as ~0.1 µmol l$^{-1}$ show that MOx rates remain high at such low concentration, both in bottom-water samples collected during oxygen-rich as well as oxygen-depleted conditions (Fig. 3, 4). This consequently provides evidence that MOB are well adapted to the seasonally sub-micromolar oxygen levels in bottom waters at Boknis Eck.



Whereas plausible explanations exist for the presence and activity of MOB under seemingly oxygen-starved conditions, we still lack an explanation for the observation that in our incubations MOx rates were highest at the lowest oxygen levels, independently of the initial oxygenation state of the sampled water (Fig. 3). The apparent adaptation to low oxygen concentrations may be part of a strategy to avoid the detrimental effects of methane starvation under oxic conditions and to

escape grazing pressure in more oxygenated water. For example, grazers were found to control the community size of MOB in shallow Finnish lakes (Devlin et al., 2015). The ability of MOB to operate at low oxygen levels apparently enables them to thrive in bottom waters with only trace amounts of oxygen. Additionally, reduced grazing activity under hypoxia may allow efficient MOB community growth, which would lead to elevated MOx activity in these water layers. Indeed, direct links between the MOB community size and MOx have been demonstrated before (e.g., Steinle et al., 2015).

Grazing within the incubations themselves is probably not responsible for the observed trend of highest MOx at low oxygen levels; particularly not in incubations conducted with originally hypoxic water (June 2014; Fig. 3b), where most grazers were likely absent from the beginning of the experiment. Enhancement of MOx rates in incubations at low oxygen levels is thus best explained at the cellular/biochemical level. Aerobic microbes produce reactive oxygen species (ROS; including superoxide anion radicals, hydrogen peroxide and hydroxyl radicals) as by-products during their metabolism, which can

cause oxidative damage to cellular structures. The amount of ROS leaking from the respiratory chain typically increases at elevated oxygen concentrations (see review by Baez and Shiloach, 2014). Although we can only speculate as to how effective MOB are able to remove ROS (e.g. by catalase), it seems possible that MOx may be slowed down at high oxygen concentrations. However, the fraction of $^{14}$C incorporated into biomass was markedly higher at elevated oxygen concentration, even though MOx rates were higher at low oxygen concentration (Fig. 4), which rather suggests a differential

metabolic functioning of MOB at low versus high oxygen concentrations. Kalyuzhnaya et al. (2013) showed that, under oxygen-deficiency stress (<10 µmol l$^{-1}$), a strain of type I MOB (*Methylomicrobium alcaliphylum* 20Z) switched to fermentative methane utilization, leading to a reduced MOB-biomass synthesis and the enhanced formation of short-chain carbonic acids (i.e., formate, acetate, succinate) as metabolic end products. If the Boknis Eck-MOB would employ a similar, energetically less favourable methane utilization pathway, our results could be explained by an overall higher catabolic

activity at the cost of anabolic investment at low oxygen concentrations. Additional biochemical investigations are necessary to constrain the metabolic partitioning of MOB at high versus low oxygen levels further, but also to test the potential role of ROS in MOx attenuation at elevated oxygen levels. Finally, we also cannot completely rule out that diverse MOB species with different oxygen requirements are active under different oxygen-concentrations in our incubations (and at Boknis Eck), which may also lead to the observed discrepancies in the MOx rate and metabolic functioning.



### 4.3 Temperature effects on MOx

#### 4.3.1 Mesophilic behaviour of MOx and implications for future warming

With the exception of Nov. 2013 (see Sect. 4.4, below), the MOB community at Boknis Eck generally showed a mesophilic behaviour, with temperature optima >10 °C and <37 °C, both in bottom (Fig. 5) and surface waters, respectively (Fig. S2).

This is consistent with temperature optimum ranges observed for most cultured aerobic methanotrophs (Hanson and Hanson 1996; Murrell 2010). The temperature optimum was ~5–20 °C higher than the in situ temperature (at the time of sampling), which is typical for many microorganisms (Price and Sowers 2004). The on-going increase in atmospheric temperature has led to a global sea surface temperature anomaly of about 0.5 °C (until 2010) and future projections indicate further warming by up to 2 °C until the middle of the 21st century (IPCC, 2013). A similar warming trend is observed for the Baltic Sea

(HELCOM, 2009). It is unclear how exactly surface water warming (and associated physico-biogeochemical side effects in the upper water column) will influence bottom water temperatures in the future. Particularly in shallow shelf seas, however, ocean surface warming is likely to propagate to deeper water layers, and indeed, such a warming trend has been recorded at Boknis Eck (Lennartz et al., 2014). Our incubation experiments indicate that the present MOB community is well adapted to temperature changes of a few degrees Celsius, and that $r_{MOx}$ will probably increase in the near future.

#### 4.3.2 Indirect evidence for a change of the MOB community by North Sea water inflow events

Contrary to the general mesophilic MOB community at Boknis Eck, incubations with water samples from Nov. 2013 (Fig. 5b, Fig. S2a) revealed a psychrophilic behaviour of the MOB with a much lower temperature optimum at about 10 °C. This difference strongly suggests the presence of another microbial MOB community at Boknis Eck in Nov. 2013 compared to the other sampling times. The sampling took place on 8 November 2013, about one week after a larger inflow event that

transported oxygen-rich, salty North Sea water into the Baltic Sea. Such input is clearly indicated by salinity anomalies observed at several hydrographic measurement stations of the Federal Maritime and Hydrographic Agency of Germany (Nausch et al., 2014), including the nearby station Kiel Lighthouse (see also the online data base of hydrographic measurement station of the German "Bundesamt für Seeschifffahrt und Hydrographie"; www.bsh.de). No continuous salinity measurements were conducted at Boknis Eck to provide evidence for the saltwater injection directly at the study site.

However, the unusual oxygen profile with a distinct minimum at 15 mbsl and increasing oxygen concentrations in bottom waters seems to attest to the October inflow event, during which dense undercurrents of salt- and oxygen-rich water from the North Sea were wedging into, and (partly) displacing, the hypoxic deep Eckernförde Bay water.

In Nov. 2013, maximum $k$ and $r_{MOx}$ were detected at 15 mbsl, and not in bottom waters, as it was the case during all the other samplings (Fig. 2). In combination with the observed shift from a meso- to a psychrophilic community, this suggests that

with the inflowing North Sea water, the local MOB community at Boknis Eck was displaced. North Sea water is characterised by elevated methane concentrations (Rehder et al., 1998) and relatively high water column MOx rates were detected at several sites in the North Sea (Mau et al., 2015; Osudar et al., 2015; Steinle et al., in press), so that the inflowing



water may indeed contain relatively high numbers of MOB. However, the inflow and the transient replacement of hypoxic Eckernförde Bay water (including its microbial stock) has led to reduced MOx rates by the imported psychrophilic compared to the autochthonous mesophilic MOB community near the sea floor. Similarly, current-associated translocation of MOx communities has been found to constitute an important control on the magnitude of local MOx rates offshore Svalbard

(Steinle et al., 2015). In Eckernförde Bay, the origin of the seemingly differential MOB communities, their actual cell numbers, and the effects of sporadic, short-term perturbations on the MOx potential need to be constrained further in future studies.

**4.4 Considerable methane removal by MOx**

The calculated median methane efflux from our measurements (5.1 or 11.9 µmol m$^{-2}$ d$^{-1}$, considering min. and max. values

for $k$, respectively) was very similar to previous data from a monthly sampling campaign at Boknis Eck between 2006 and 2008 (6.3–14.7 µmol m$^{-2}$ d$^{-1}$; Bange et al., 2010). Average surface saturation (447% with respect to atmospheric equilibrium) at Boknis Eck was at the lower end of European estuarine systems and river plumes, but clearly higher than values determined for European shelf waters (Upstill–Goddard et al., 2000; Bange, 2006; Schubert et al., 2006; Grunwald et al., 2009; Ferrón et al., 2010; Schmale et al., 2010; Osudar et al., 2015; Upstill–Goddard and Barnes, 2016). At Boknis Eck,

methane originates from sedimentary methanogenesis (Whiticar 2002; Treude et al., 2005; Maltby 2015), which can enter the water column either by diffusion or by bubble transport. The amplitudes of methanogenesis and AOM were found to show strong spatiotemporal heterogeneity, which makes flux estimates difficult (Treude et al, 2005; Maltby 2015). Additionally, methane input into the water column via bubble transport has not been assessed quantitatively thus far at Boknis Eck. As a result, we can only speculate on the relative importance of methane release from sediments by diffusion

versus ebullition on the total methane flux. Here, we assume that the release of methane from sediments to the water column equals the total flux of methane ($F_{sed} = F_{tot}$), i.e., the sum of methane oxidized via MOx in the water column and the evasion of the remaining methane to the atmosphere ($F_{atm} + F_{MOx} = F_{tot}$). Possible effects of advection on the methane concentration and MOx are ignored. With these simplifications, we can estimate the fraction of water column methane removed by MOx before its potential emission into the atmosphere. Our depth-integrated rates and the average of our minimum and maximum

estimates of methane efflux to the atmosphere imply that 71–95% of the methane released from the sediment to the water column is oxidized during stratified conditions (Table 2), underscoring the high efficiency of the water column methane filter during summer/fall, albeit the shallow water depth. In contrast, only 39–58% is oxidized during mixed conditions during winter/spring. In combination with a generally higher (community-driven) MOx potential, the limited exchange between bottom waters and the upper mixed layer under stratified conditions, and hence a lower turbulent diffusive methane

flux, are conducive to efficient methane oxidation and contribute to the higher MOx filter capacity during summer/fall. In conclusion, seasonal water column stratification thus not only promotes generally higher rates of MOx but also has a modulating effect on the efficiency of the microbial methane filter in the water column.



## 5. Conclusions and implications

To the best of our knowledge, this is the first long-term (>2 years) seasonal study on MOx activity in a seasonally stratified/hypoxic coastal environment. The results are important for the understanding of the temporal dynamics and environmental controls on the efficiency of the MOx methane filter in shallow marine environments. We demonstrate that

methane oxidation at Boknis Eck is strongly affected by changes of water column stratification, as well as by inflow events from the North Sea, and the associated episodic displacement of bacterial communities with different MOx potential. Moreover, we provide evidence that MOB at Boknis Eck are well adapted to very low oxygen concentrations, with a clear tendency of highest MOx rates occurring in the sub-micromolar range. Although the exact ecological driving forces, as well as the biochemical mechanisms behind this adaptation remain uncertain, it seems likely that the capacity to thrive in low-

oxic waters enables MOB to evade grazing pressure in the more oxygenated parts of the water column.

Our field and laboratory investigations revealed that very low oxygen concentrations under stratified conditions and at elevated temperatures are particularly conducive to high MOx rates and efficient methane "removal" from the water column. Ongoing trends at Boknis Eck (and in many other coastal ecosystems) predict warmer temperatures in the future, and probably an earlier onset of seasonal stratification. However, for a future scenario, it remains unclear as to what extent the

low-oxygen adaptation and a temperature-related enhancement of MOx will counterbalance elevated rates of methanogenesis in the sediments caused by higher temperatures. Our study suggests that an extension of the hypoxic period and increasing temperatures will not necessarily lead to higher methane evasion to the atmosphere. Ultimately, the oxygen–MOx link will also depend on the potential inhibition of MOB growth under oxygen-depleted conditions. Our experimental data suggest that the MOB communities can experience growth restrictions under oxygen-deficiency stress, but whether the expected

spatiotemporal expansion of hypoxia, and eventually anoxia, may finally hamper MOx has to be investigated further.

### Author contribution

L. Steinle, J. Maltby, T. Treude, M. F. Lehmann, and H. Niemann designed the study. L. Steinle and J. Maltby carried out onboard sampling. L. Steinle, J. Maltby, A. Kock, and H. W. Bange conducted further geochemical analyses. L. Steinle measured microbial rates and performed incubation experiments. H. Niemann helped with incubation experiments. L. Steinle

prepared the manuscript with contributions from all authors.

The authors declare that they have no conflict of interest.

### Acknowledgements

We thank the captains and crews of R/V Alkor, R/C Littorina and R/B Polarfuchs, and the staff of the GEOMAR's

Technology and Logistics Centre for the excellent support at sea and onshore. Additional thanks go to G. Schüssler, F.





Wulff, P. Wefers, A. Petersen, M. Lange, and F. Evers for help with the fieldwork. We also thank F. Malien, X. Ma, S. Lennartz and T. Baustian for the regular calibration of the CTD and $CH_4$ analysis. This work received financial support through a D–A–CH project funded by the Swiss National Science Foundation and the German Research foundation (grant no. 200021L_138057, 200020_159878/1), and through the Cluster of Excellence "The Future Ocean" funded by the German

Research Foundation. Further support was provided through the EU COST Action PERGAMON (ESSEM 0902).

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



**Tables and Figures**

**Table 1. Overview of sampling dates and sampled parameters. "temp. dep." indicates samples used for temperature dependency incubations, "O₂ dep. " indicates samples used for O₂ incubations with ³H-CH₄ and ¹⁴C-MOx for O₂ incubations with ¹⁴C-CH₄ and determination of ¹⁴C-incorporation into biomass.**

| sampling | Date | $CH_4$ | $r_{MOx}$ | temp. dep. | | O₂ dep. | | ¹⁴C-MOx | |
|----------|------|--------|-----------|------------|--------|---------|--------|---------|---------|
| | | | | 5 mbsl | 20 mbsl | 5 mbsl | 20 mbsl | 5 mbsl | 20 mbsl |
| Oct. '12 | 17.10.12 | x | x | | | | | | |
| Mar. '13 | 13.03.13 | x | x | | | | | | |
| June '13 | 27.06.13 | x | x | | | | | | |
| Sept. '13 | 05.09.13 | x | x | | x | | | | |
| Nov. '13 | 08.11.13 | x | x | x | x | | x** | | x |
| Feb. '14 | 24.02.14 | x | x | x | x* | x | x* | x | x* |
| June '14 | 18.06.14 | x | x | x | x | x | x | x | x |
| Sept. '14 | 17.09.14 | x | x | x | x | | | | |

\* because of disturbance of the water column, we used water from 25 mbsl instead of 20 mbsl

\*\* because of disturbance of the water column, we used water from 15 mbsl instead of 20 mbsl

**Table 2. Integrated methane oxidation rates ($F_{MOx}$) and methane flux into the atmosphere ($F_{atm}$) in comparison to stratification (indicated by the buoyancy frequency N). $F_{atm}$ was calculated with a min and max $k_w$-value. $F_{tot} = F_{MOx} + F_{atm-avg}$. N is given as the average for 10–20 mbsl. A water column with N values >120 is considered stratified.**

| sampling | $F_{MOx}$ [µmol l⁻¹ d⁻¹] | $F_{atm}$ - min $k_w$ [µmol l⁻¹ d⁻¹] | $F_{atm}$ - max $k_w$ [µmol l⁻¹ d⁻¹] | $F_{MOx}/F_{tot}$ [%] | N [cph] | stratification |
|----------|------|------|------|------|------|------|
| Oct. '12 | 29.6 | 12.4 | 31.1 | 58 | 91 | weak |
| Mar. '13 | 11.7 | 10.5 | 26.4 | 39 | 59 | weak |
| June '13 | 27.3 | 3.2 | 7.9 | 83 | 241 | strong |
| Sept. '13 | 33.0 | 3.4 | 8.5 | 85 | 229 | strong |
| Nov. '13 | 28.0 | 6.3 | 15.8 | 72 | 216 | strong |
| Feb. '14 | 14.5 | 7.0 | 17.6 | 54 | 114 | weak |
| June '14 | 12.7 | 2.9 | 7.4 | 71 | 201 | strong |
| Sept. '14 | 82.3 | 2.5 | 6.2 | 95 | 245 | strong |



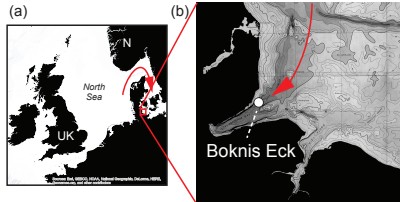

**Figure 1: Map of the North- and the Western Baltic Sea with a close-up of the study area. (a), Overview map of the Western Baltic Sea including the Kattegat, the Skagerrak and parts of the North Sea. (b), close-up of the study area with the sampling site (time-series station Boknis Eck) marked with a white dot. Red arrows indicate sporadic inflow of North Sea water.**

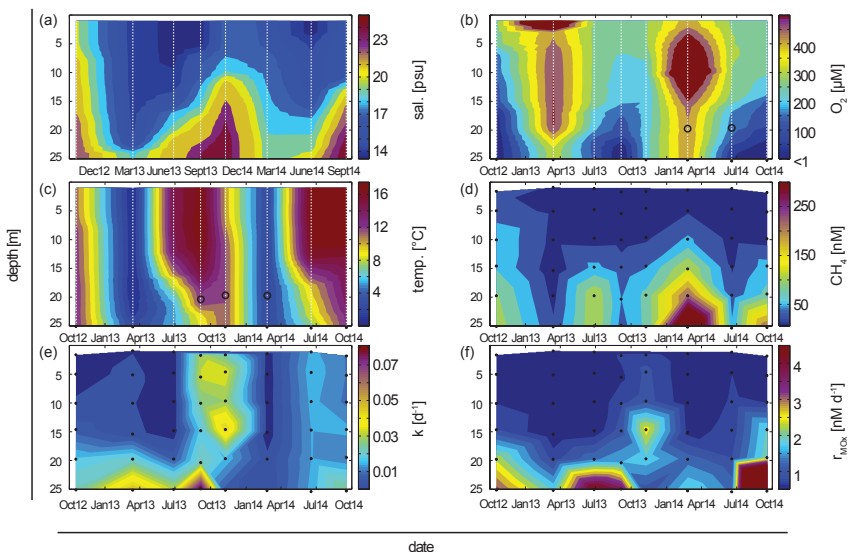

**Figure 2. Seasonal variability of physico-chemical parameters and aerobic methanotrophy at the Boknis Eck station. Distribution**
10 **of salinity (a), oxygen (b), temperature (c), methane (d), first order rate constants of aerobic methane oxidation *k* (e), and aerobic methane oxidation rates *r_MOx* (f). Positions of discrete samples (dots) and continuous measurements (dashed lines) are indicated. Depths of water used for oxygen and temperature incubation experiments (Fig. 3, 4 and 5) are indicated with circles in (b) and (c).**





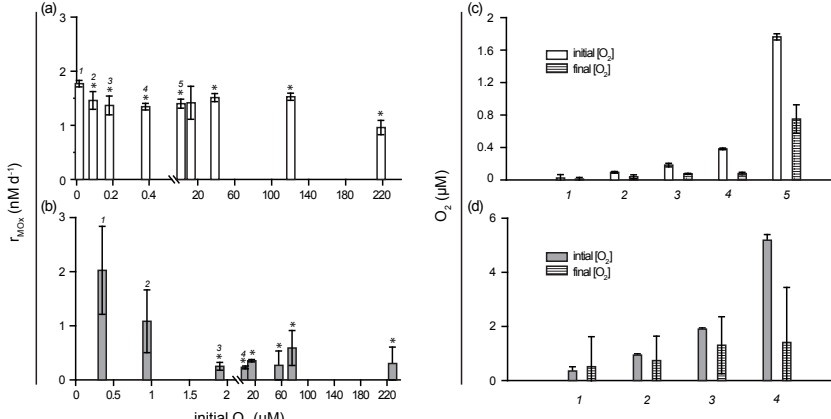

**Figure 3. MOx rates in incubations with adjusted oxygen concentrations.** $r_{MOx}$ determined in incubations of Boknis Eck water (20 mbsl) with $^3$H-CH$_4$ in Feb. 2014 (a) and June 2014 (b). Asterisks indicate a p-value <0.05 of a two-tailed, two-sample t-test assuming equal variance of the MOx rate compared to the MOx rate of the lowest oxygen concentration. Corresponding oxygen concentrations determined at the beginning and the end of incubations from the Feb. 2014 (c) and the June 2014 sampling (b) for

5  incubations with oxygen concentration <10 µmol l$^{-1}$. Incubations were performed in triplicates and standard deviations are indicated. For oxygen concentrations >10 µmol l$^{-1}$, oxygen concentrations at the end of incubations were 14–40% (Feb. 2014) and 3–8% (June 14) lower than at the beginning (data not shown).

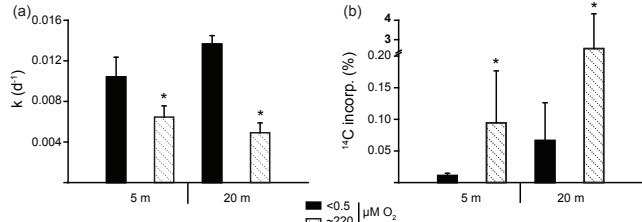

**Figure 4. Methane-carbon assimilation in relation to oxygen concentration.** Methane-carbon assimilation was determined from incubations amended with $^{14}$C-CH$_4$ at saturated (~220 µmol l$^{-1}$, shaded bars) or low oxygen concentrations (<0.5 µmol l$^{-1}$, black bars) of water recovered in June 2014 at 5 mbsl and 20 mbsl. Incubations were performed in triplicates and standard deviations are indicated. (a) First-order rate constant ($k$). (b) Fraction of oxidized methane incorporated into biomass. Asterisks indicate a p-

15  value <0.05 of a two-tailed, two-sample t-test assuming equal variance between the samples at low and high oxygen concentrations.





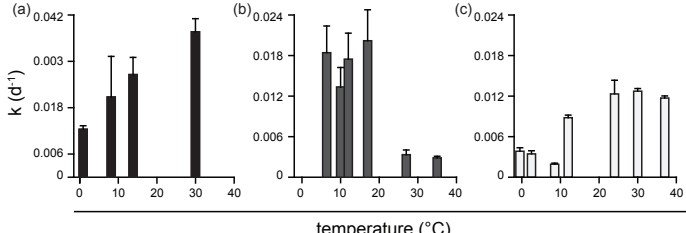

**Figure 5. MOx first order rate constants ($k$) at different temperatures. First order rate constants of MOx were determined with $^{3}$H-CH$_4$ amendments in triplicated incubations of water from 20 mbsl recovered in Sept. 2013 (a), Nov. 2013 (b), and Feb. 2014 (c).**