# Peer review of "Effects of low oxygen concentrations on aerobic methane oxidation in seasonally hypoxic coastal waters"

_Biogeosciences, 2016_

## Referee Comment (RC1) · S. Mau (Referee) · 6 Dec 2016

Steinle et al. found that oxygen concentrations and temperatures control the rate of aerobe methane oxidation in coastal waters. The authors sampled the water column at a station with regular hypoxic conditions every 2-5 month over a time frame of two years. They measured physico-chemical water properties, methane concentrations, and methane oxidation rates (MOx). They observed that although methane concentrations are not linked to the different seasons in the bay, MOx related to the seasons. Higher MOx were measured in the summer/fall when the water column was stratified in contrast to lower MOx in winter/spring when the water column was mixed. The higher rates correlate with higher temperatures and lower oxygen concentrations during the

stratification period and the lower rates relate to lower temperatures and higher oxygen concentrations in a mixed water column. Experiments were implemented to validate the correlations. Highest MOx-rates were measured at a temperature of ∼30°C, except when North Sea inflow water was sampled, then highest MOx-rates were measured at ∼10°C. The experiments further showed that methane oxidation rates were elevated at lower oxygen concentrations compared to higher oxygen concentrations, with increased biomass incorporation at high oxygen concentrations. Both field data and experiments thus indicate a negative correlation between oxygen concentrations and methane turnover.

Although culture studies of known methanotrophic bacteria have already shown the influence of oxygen and temperature on microbial methane oxidation as summarized by Hanson and Hanson (1996, Microbio Rev, 60, 2, 439-471), investigations of these relationships in the marine realm lag behind. Steinle et al. thus target a severe knowledge cap in the marine methane cycle. The authors present a comprehensive, well written study that comprises a rather long time series and a combination of field data with experiments. While the former should be highlighted as it involves a lot of logistical work, the latter should be stressed as a solid scientific approach to analyze field data and evaluate the results by implementing laboratory experiments. Therefore, I recommend publication of the manuscript after considering my few comments and remarks.

I start with some major remarks:

1. Section 4.2 of the discussion: I don't understand why the authors do only shortly discuss different methanotrophic bacteria as the source of higher/lower methane turnover. It is known that type II methanotrophs can utilize methane better at low oxygen concentrations whereas type I methanotrophs utilize methane at higher oxygen concentrations (Amaral and Knowles, 1995). There is also a difference of biomass incorporation between these two types; type II can assimilate much higher portions of $CO_2$ as carbon source (up to 50%) than type I methanotrophs (up to 15%) (see reference in Strong et al., 2015, Environmental Science and Technology, 49, 4001-4018). Taking both

aspects together could explain the different assimilation patterns and turnover rates obtained.

2. Section 4.3.1 of the discussion: By using the temperature dependence, you can calculate the Q10 factor and compare with the one derived in Bussmann et al., 2015 (L&O Methods, 13, 312-327). Taking the Q10, you can further evaluate the in- or decrease of MOx-rates due to temperature and compare those with your field data. If you know the MOx-rate change due to temperature, you can differ between the temperature and oxygen effect in your field data. It would be great to know if temperature or oxygen has a stronger effect on methane turnover.

3. Section 4.4 of the discussion: The fraction of MOx (FMox/Ftot in %) is high during stratification, because Fatm is low, not because of higher MOx-rates (attached Fig. 1). Water column stratification clearly affects the sea-air flux (right graph of Fig. 1). I see a better distinction between the seasons by solely comparing FMOx. If I list FMOx from lowest to highest, I get: spring 11.7, summer 12.7, winter 14.5, summer 27.3, fall 28, fall 29.6, fall 33, fall 82.3 $\mu$mol/m2d, that is FMOx is always higher in fall compared to the other seasons. As this already illustrates that 'MOx exhibits a seasonal variability', why do you calculate the MOx-fraction of the total loss terms of methane (FMOx + Fatm)? I recommend to stick to your results of FMox and Fatm and delete the Ftot assumption as the flux of methane from the ground was not measured. Ignoring dispersion and advection of methane in the water column might be an oversimplified view, which can easily produce wrong results.

4. Figure 4: Do you have any explanation, why so less carbon was assimilated? I know ratios of biomass to CO2 of 0.12-0.4. Your results appear much lower.

Minor remarks:

Page 1, line 21: Please change 'always' to 'generally' as you write later in the manuscript that in Nov. 2013 MOx-rates were higher in the water column than just above the ground.

Page 2, line 6: Here you could use an 'old' reference in addition, which describes the degradation of organic matter by methanogenic archaea, otherwise it seems like it is a newly discovered process.

Page 4, line 17: The use of mercury chloride solution should be avoided as it is very toxic. In addition, mercury chloride can be transformed to less toxic substances by methanotrophs and thus might not poison all methanotrophs (Boden and Murrell, 2011, FEMS Microbiol Lett 324, 106–110).

Page 5, line 25: It would be nice to include the difference of the temperature of the experiments to the in situ temperature in this section, otherwise the reader has to search the text for it.

Page 6, line 11: The value 600 is the Sc of $CO_2$ at 20°C in freshwater. The Sc of $CH_4$ is slightly different: 617 at 20°C in freshwater. Wanninkof, 2014 (L&O Methods, 12, 351-362) forwards an equation to derive Sc for both seawater and freshwater from temperature. Please include an error evaluation, what effect does this change have.

Page 6, line 17: The numbers of kW are for $CO_2$, please state this in the text.

Page 8, line 2, 3, 5: Why is there a difference in oxygen concentration between 3H-$CH_4$ and 14C-$CH_4$ experiments? For 3H-$CH_4$ experiments you differ between above and below 15 $\mu$mol/L, but for 14C-$CH_4$ you differ between above and below 0.5 $\mu$mol/L.

Page 9, line 28: Are the results of the pearson and two-tail's student test derived for the relationship k v. $CH_4$ or MOx v. $CH_4$? I assume it is for the relation k-$CH_4$, could you include the test results for MOx-$CH_4$, too.

Page 12, line 25: This unusual oxygen profile is not visible in Fig. 2.

Tab. 2: The units are wrong. It should be flux units per unit square area ($\mu$mol m-2 d-1), not per volume.

Overall, the scientific outcome of the manuscript is sound and will remain; I suggest only to add some valuable details. I enjoyed reading the manuscript and look forward to its final publication.

[Figure]

[Figure]

Fig. 1: Plots derived from your data in Table 2.

**Fig. 1.** Plots derived from your data in Table 2

---

## Referee Comment (RC2) · Anonymous Referee #2 · 9 Dec 2016

The study from Steinle et al on the "Effects of low oxygen concentration on aerobic methane oxidation in hypoxic coastal waters" is well written and very interesting. I especially appreciate the experimental set-up, as it is certainly very tricky to perform incubations at defined $O_2$ and $CH_4$ concentrations! However, I have some questions and remarks (most of them are typed into the pdf file). I think it is a bit confusing that the authors jump between the usage of k' and MOX. Sometimes they report on corelation for MOX and some times corelations for k. This should be handled in a more coherent way. The statistic tests should be explained in more detail in the M&M section, and not in the figure legends. I also think that there are more informations hiden in the in situ data. For example one could split the data set into surface and

bottom water and than do seperate statistics. In addition I thought it is good / common practice now that at least the in situ data should be made available for the public. I could not find any indication here!

Please also note the supplement to this comment:
http://www.biogeosciences-discuss.net/bg-2016-422/bg-2016-422-RC2-supplement.pdf
* * *
[Figure]

**Supplement:**

[revised manuscript text omitted]

 **Map of the North- and the Western Baltic Sea with a close-up of the study area. (a), Overview map of the Western Baltic Sea including the Kattegat, the Skagerrak and parts of the North Sea. (b), close-up of the study area with the sampling site (time-series station Boknis Eck) marked with a white dot. Red arrows indicate sporadic inflow of North Sea water.**

[Figure]

date

 **Seasonal variability of physico-chemical parameters and aerobic methanotrophy at the Boknis Eck station. Distribution**
10 **of salinity (a), oxygen (b), temperature (c), methane (d), first order rate constants of aerobic methane oxidation $k$ (e), and aerobic methane oxidation rates $r_{MOx}$ (f). Positions of discrete samples (dots) and continuous measurements (dashed lines) are indicated. Depths of water used for oxygen and temperature incubation experiments (Fig. 3, 4 and 5) are indicated with circles in (b) and (c).**

[Figure]

**Figure 3. MOx rates in incubations with adjusted oxygen concentrations.** *r*MOx determined in incubations of Boknis Eck water (20 mbsl) with $^3$H-CH$_4$ in Feb. 2014 (a) and June 2014 (b). Asterisks indicate a p-value <0.05   MOx rate compared to the MOx rate of the lowest oxygen concentration. Corresponding oxygen concentrations determined at the beginning and the end of incubations from the Feb. 2014 (c) and the June 2014 sampling (b) for incubations with oxygen concentration <10 μmol l$^{-1}$. Incubations were performed in triplicates and standard deviations are indicated.

[Figure]

**Figure 4. Methane-carbon assimilation in relation to oxygen concentration.** Methane-carbon assimilation was determined from incubations amended with $^{14}$C-CH$_4$ at  (~220 μmol l$^{-1}$, shaded bars) or low oxygen concentrations (<0.5 μmol l$^{-1}$, black bars) of water recovered in June 2014 at 5 mbsl and 20 mbsl. Incubations were performed in triplicates and standard deviations are indicated. (a) First-order rate constant (*k*). (b) Fraction of oxidized methane incorporated into biomass. Asterisks indicate  between the samples at low and high oxygen concentrations.

[Figure]

**Figure 5. MOx first order rate constants (*k*) at different temperatures. First order rate constants of MOx were determined with**
$^3$**H-CH$_4$ amendments in triplicated incubations of water from 20 mbsl recovered in Sept. 2013 (a), Nov. 2013 (b), and Feb. 2014 (c).**

---

## Referee Comment (RC3) · D. Rush (Referee) · 13 Dec 2016

This paper by Steinle et al. examined the effect of O2 concentration on aerobic methane oxidation rates at a 2-year time-series station in the Baltic Sea. They found that CH4 oxidation rates increased with increasing water depth; methane oxidation rates were highest when O2 concentrations were lowest and water temperatures highest. Overall, I found this to be an appropriate paper for Biogeosciences. It addresses the eventuality of increased marine methane release due to climate change and how potential simultaneous changes in physico-chemical parameters (e.g. increasing temperatures, decreased O2 availability) might possibly create a positive feedback to quench atmospheric methane release, in at least the Baltic Sea. However, I was left

frustrated that the authors continuously postulated hypothetically on their results without delving into actually exploring them in their discussion. For example, the simple act of investigating the aerobic methanotrophic community structure in their samples would allow them to do more than speculate about (1) temperature effects and water inflow for the North Sea causing certain time events to have dissimilar methane oxidation rates, (2) temperature optima for different aerobic methanotrophs, and (3) different metabolic functions of different communities. Was there a reason the genetics were not performed? At the very least, why not investigate the fatty acid content of these experiments to see if there is indeed a shift in functioning and/or community. I feel that the manuscript would be greatly improved with methanotroph community and biomarker data. However, if there is a valid reason for the lack of community data or these data will appear in a future paper, and the text is revised to explain this, I believe the paper is publishable with minor revisions, below.

Specific comments: There seems to be a lack of consistent acronym for aerobic methane oxidation in our community. I feel that the acronym chosen here (MOx) is not specific enough to aerobic methane oxidation. Perhaps AMOx could be used instead?

Fig.1 I'm not sure if it's just my pdf version but the figure is incredibly small and it could be more detailed

Page 4 line 16: change determinations to measurements

Page 4 line 26: what is the in situ temperature? Did it change seasonally?

Page 6 line 3: what temperatures exactly?

Page 10 line 32: insert "in the Baltic Sea" between "evidence that MOB" and "are well adapted".

Figure caption for Fig 3: panel (d) is mislabelled (b). Alternatively to 3c and 3d, a table with initial and final O2 concentrations for both sets of experiments might be more

informative?

---

## Author Response (AR1)

Replies to reviewer comments to bg-2016-422

Reply: First, we would like to thank Dr. Susan Mau, Dr. Darcy Rush, and the anonymous Reviewer for their comments that will definitely improve the manuscript. Generally, all three Reviewers wrote that our MS presents interesting new data (*"The authors present a comprehensive, well written study that comprises a rather long time series and a combination of field data with experiments." "I especially appreciate the experimental set-up, as it is certainly very tricky to perform incubations at defined $O_2$ and $CH_4$ concentrations!"*). Reviewers 1 and 3 agreed that if the questions/suggestions they had would be incorporated, the MS would be publishable with minor revisions (*"Therefore, I recommend publication of the manuscript after considering my few comments and remarks." "…I believe the paper is publishable with minor revisions, below"*), whereas Reviewer 2 did not specify recommendations for the extent of revisions (i.e. minor/moderate/major). Below, we included a point-by-point reply to the comments/concerns of all three Reviewers.

Comments by Reviewer 1, Dr. Susan Mau:
*1. Section 4.2 of the discussion: I don't understand why the authors do only shortly discuss different methanotrophic bacteria as the source of higher/lower methane turnover. It is known that type II methanotrophs can utilize methane better at low oxygen concentrations whereas type I methanotrophs utilize methane at higher oxygen concentrations (Amaral and Knowles, 1995). There is also a difference of biomass incorporation between these two types; type II can assimilate much higher portions of $CO_2$ as carbon source (up to 50%) than type I methanotrophs (up to 15%) (see reference in Strong et al., 2015, Environmental Science and Technology, 49, 4001-4018). Taking both aspects together could explain the different assimilation patterns and turnover rates obtained.*

Reply: Interesting point! That would imply, however, that type II methanotrophs should be active at low oxygen concentrations, and then assimilate higher portions of $CO_2$ as a carbon source than type I methanotrophs, which would be active at higher $O_2$ concentrations. Yet, we observed the opposite relationship: at low oxygen concentrations, less carbon was incorporated, so that our observations cannot be explained by a switch-over from type I to type II methanotrophy. Additionally, unpublished data from another study we carried out in Eckernförde Bay show that there are mostly type I MOB present, also arguing against this hypothesis. We will discuss this hypothesis in the revised version of the MS.

*2. Section 4.3.1 of the discussion: By using the temperature dependence, you can calculate the Q10 factor and compare with the one derived in Bussmann et al., 2015 (L&O Methods, 13, 312-327). Taking the Q10, you can further evaluate the in- or decrease of MOx-rates due to temperature and compare those with your field data. If you know the MOx-rate change due to temperature, you can differ between the temperature and oxygen effect in your field data. It would be great to know if temperature or oxygen has a stronger effect on methane turnover.*

Reply: Thank you for this suggestion. We tried it, and obtained values between 1.4 and 2.2 for Q10 calculated according to Bussmann et al. (2015) for Sept. 2013 and Feb. 2014. In the recent literature, the use and comparability of Q10 is discussed. Alster et al. (2016, Front. Microbiol.), for instance, write: "Q10 gives a false sense that a single constant can characterize the temperature sensitivity of a system (Davidson and Janssens, 2006). In order to overcome this obvious discrepancy authors using Q10 often present multiple temperature sensitivity values at different temperature ranges for a given system, leading to results that are often difficult to compare." We think that the data set we obtained in our experiments contains insufficient data points (4-6) per date and depth in order to calculate robust/reliable Q10 values. Already the simple inspection of our data set allows a semi-quantitative assessment of the higher relative importance of $O_2$ versus T in regulating MOx. This will be clarified in the revised MS.

*3. Section 4.4 of the discussion: The fraction of MOx (FMox/Ftot in %) is high during stratification, because Fatm is low, not because of higher MOx-rates (attached Fig. 1). Water column stratification clearly affects the sea-air flux (right graph of Fig. 1). I see a better distinction between the seasons by solely comparing FMOx. If I list FMOx from lowest to highest, I get: spring 11.7, summer 12.7, winter 14.5, summer 27.3, fall 28, fall 29.6, fall 33, fall 82.3 _mol/m2d, that is FMOx is always higher in fall compared to the other seasons. As this already illustrates that 'MOx exhibits a seasonal variability', why do you calculate the MOx-fraction of the total loss terms of methane (FMOx + Fatm)? I recommend to stick to your results of FMox and Fatm and delete the Ftot assumption as the flux of methane from the ground was not measured. Ignoring dispersion and advection of methane in the water column might be an oversimplified view, which can easily produce wrong results.*

Reply: Thank you very much for this valuable input. We agree that we simplify the system, ignoring dispersion and advection of methane. We will revise the MS according to your comments (i.e., take out the assumption of Ftot=FMOx+Fatm). Reviewer 2 suggested to calculate total methane content of a given date and to relate the estimate to FMOx and Fatm, which we will do in the revised version of the MS. With regard to the second part of your comment: it is true, indeed, that Fatm is always lower when stratification is higher. In this context, we will add the figure you attached to your review made with our data in the supplement. The point we wanted to make, though, is that without any methane oxidation, methane concentrations would reach very high levels. Our assumption was that Fatm is low mainly because methane is trapped more efficiently and for a longer period of time underneath the thermocline/density gradient, and hence there is more time for methane oxidation to proceed. We will clarify this in the updated MS.

*4. Figure 4: Do you have any explanation, why so less carbon was assimilated? I know ratios of biomass to CO2 of 0.12-0.4. Your results appear much lower.*

Reply: We think that the ratio of carbon incorporated to $CH_4$ turnover is highly dependent on the environmental conditions. Especially in low-oxygen experiments, methanotrophs appear to encounter sub-optimal conditions, where growth is limited. Similarly, high sulfate turnover rates were measured sulfate reducer assays, but growth could almost not be detected under sub-optimal conditions (Cypionka 2000, Annu. Rev. Microbiol.). In some of the incubations, the measured C-incorporation/MOx ratios reached up to 0.09, which is not far from the range you mentioned.

*Minor remarks:*
*Page 1, line 21: Please change 'always' to 'generally' as you write later in the manuscript that in Nov. 2013 MOx-rates were higher in the water column than just above the ground.*

Reply: We will do that.

*Page 2, line 6: Here you could use an 'old' reference in addition, which describes the degradation of organic matter by methanogenic archaea, otherwise it seems like it is a newly discovered process.*

Reply: We will do that.

*Page 4, line 17: The use of mercury chloride solution should be avoided as it is very toxic. In addition, mercury chloride can be transformed to less toxic substances by methanotrophs and thus might not poison all methanotrophs (Boden and Murrell, 2011, FEMS Microbiol Lett 324, 106–110).*

Reply: Thank you for this suggestion. We already changed to using NaOH to fix the samples in our lab. We are aware of the potential biasing effects of using HgCl in MOx incubations.

However, in the experiments presented in this publication, the killed controls were about 1% of the average tracer turnover in our experiments, and we can hence conclude that most methanotrophs were poisoned. An advantage of using HgCl2 is that it does not alter the solubility of gases as much as the addition of high-concentrate NaOH.

*Page 5, line 25: It would be nice to include the difference of the temperature of the experiments to the in situ temperature in this section, otherwise the reader has to search the text for it.*

Reply: That's a very good suggestion, which we will follow.

*Page 6, line 11: The value 600 is the Sc of $CO_2$ at 20°C in freshwater. The Sc of $CH_4$ is slightly different: 617 at 20°C in freshwater. Wanninkof, 2014 (L&O Methods, 12, 351-362) forwards an equation to derive Sc for both seawater and freshwater from temperature. Please include an error evaluation, what effect does this change have.*

Reply: The kw value was taken from Raymond and Cole (2001) as k600 for $CO_2$. To this end, we needed to correct kw for $CH_4$ by multiplying it with $(ScCH_4/600)^{-0.5}$, where $ScCH_4$ is the Sc number for $CH_4$ computed for the temperature and salinity at the time of the measurements. The equation(s) for the Sc number given by Wanninkhof (2014) are valid only for sal 35 (open ocean) and 0 (freshwater). Please note that the salinities in the Eckernförde Bay range from 12-24 (see Lennartz et al., Biogeosci., 2014) and, therefore, the Wanninkhof equations are not applicable.

*Page 6, line 17: The numbers of kW are for CO2, please state this in the text.*

Reply: We modified equation (4) which reads now: Fatm = k600 $(ScCH_4/600)^{-0.5}$ x ($[CH_4]$ – $[CH_4]eq$) and state in the text 'Following the recommendation by Raymond and Cole (2001) for coastal systems, we used 3 cm h-1 ( = $0.83*10^{-6}$ m s-1) and 7 cm h-1 (=$1.94*10^{-5}$ m s-1) as minimum and maximum values for k600, respectively.'
Please note that the originally cited values were erroneously cited from Bange et al. (2010), as they report a kw value that already includes the Sc correction.

*Page 8, line 2, 3, 5: Why is there a difference in oxygen concentration between [3]H-$CH_4$ and [14]C-$CH_4$ experiments? For [3]H-$CH_4$ experiments you differ between above and below 15 µmol/L, but for [14]C-$CH_4$ you differ between above and below 0.5 µmol/L.*

Reply: The experiments with [3]H-$CH_4$ were performed with a range of different oxygen concentrations (Figure 3), whereas the experiments with [14]C-$CH_4$ were only carried out with two different oxygen concentrations (saturated and below 0.5 umol/l). We will add some clarifications in the method section.

*Page 9, line 28: Are the results of the pearson and two-tail's student test derived for the relationship k v. $CH_4$ or MOx v. $CH_4$? I assume it is for the relation k-$CH_4$, could you include the test results for MOx-$CH_4$, too.*

Reply: The results included in the MS are for k versus $CH_4$. The results for MOx versus $CH_4$ show qualitatively exactly the same, but with different numbers. We will include test results for MOx-$CH_4$ in the updated version. Reviewer 2 also asked for clarification/streamlining of different statistical tests carried out with k and/or MOx versus other parameters. We agree that it was not always clear, and we will improve this part in the revised version of the MS.

*Page 12, line 25: This unusual oxygen profile is not visible in Fig. 2.*

Reply: This is indeed difficult to see in the contour plot. We will add a profile plot of the

Reply: November 2013-sampling to the supplementary section.

*Tab. 2: The units are wrong. It should be flux units per unit square area (_mol m-2 d-1), not per volume.*

Reply: Thank you very much for noticing. We will correct this.

Answers to Reviewer 2:
*The study from Steinle et al on the "Effects of low oxygen concentration on aerobic methane oxidation in hypoxic coastal waters" is well written and very interesting. I especially appreciate the experimental set-up, as it is certainly very tricky to perform incubations at defined $O_2$ and $CH_4$ concentrations! However, I have some questions and remarks (most of them are typed into the pdf file).*
*I think it is a bit confusing that the authors jump between the usage of k' and MOX. Sometimes they report on corelation for MOX and some times corelations for k. This should be handled in a more coherent way.*

Reply: We will make sure to be consistent in an updated version of the manuscript.

*The statistic tests should be explained in more detail in the M&M section, and not in the figure legends.*

Reply: We will add a paragraph about the statistics in the M&M section, but we also want to keep it in the figure legends.

*I also think that there are more informations hidden in the in situ data. For example one could split the data set into surface and bottom water and than do separate statistics.*

Reply: We tried this before, but did not gain additional information from the data set. We will upload our data on PANGAEA so the data is available for independent analyses by other research groups.

*In addition I thought it is good / common practice now that at least the in situ data should be made available for the public. I could not find any indication here!*

Reply: See comment above.

Reply: In addition to these general comments, Reviewer 2 had several minor remarks and suggestions for improving the manuscript marked in an annotated pdf file. We will follow essentially all suggestions. We added our replies to the comments directly in the pdf file. Some of the more important remarks marked in the pdf include the use of the most recent equation to calculate methane fluxes and re-evaluate how we calculate the fluxes and methane reservoirs in the water column. For this, please also see our comments in the pdf file, as well as our response to Reviewer 1. Additionally, Reviewer 2 asked for more specifications on the experimental procedures of the oxygen manipulation experiments, which we will provide. Finally, Reviewer 2 also asked us to calculate Q10 based on data from our temperature incubations. For this, see our response to above.

Answers to Reviewer 3, Dr. Darci Rush:
*…. I was left frustrated that the authors continuously postulated hypothetically on their results without delving into actually exploring them in their discussion. For example, the simple act of investigating the aerobic methanotrophic community structure in their samples would allow them to do more than speculate about (1) temperature effects and water inflow*

*for the North Sea causing certain time events to have dissimilar methane oxidation rates, (2) temperature optima for different aerobic methanotrophs, and (3) different metabolic functions of different communities. Was there a reason the genetics were not performed? At the very least, why not investigate the fatty acid content of these experiments to see if there is indeed a shift in functioning and/or community. I feel that the manuscript would be greatly improved with methanotroph community and biomarker data. However, if there is a valid reason for the lack of community data or these data will appear in a future paper, and the text is revised to explain this, I believe the paper is publishable with minor revisions, below.*

Reply: Reviewer 3 felt that the manuscript would be improved by additional genetic- and/or biomarker analysis. While this is, in principal, a very good suggestion, such analyses would go far beyond the scope of our manuscript. The main goal was not to investigate the methanotrophic community, but rather seasonal changes in methane-oxidation potential and biogeochemical controls. We did not carry out biomarker analysis of seawater samples because detection of methanotrophs, or even more specifically, constraining changes in the methanotrophic community is not straightforward based on biomarker analyses alone, as lipid specificity is limited, and since methanotrophs represent typically only a minor fraction of the marine microbial community. Hence, methanotrophic biomarker signatures may be masked by more abundant lipids.
Indeed, in order to detect changes of the methanotrophic communities, a (work-intensive) molecular approach (i.e., NGS, clone libraries, qPCR) would have helped. Yet, again, this would have exceeded the framework of our MS. Similarly, in order to investigate the different metabolic functioning of communities at low and high oxygen concentrations, studies aiming at tracing the methane-carbon inside the methanotrophs would have to be conducted, which we are currently working on.
The discussion on possible shifts in the microbial community only plays a minor role in the manuscript, and we agree that this part of the paper remains rather speculative since we do not provide any genetic analysis. Nevertheless, as a starting point for future work, it seems worthwhile to cautiously state that shifts in the temperature optimum may be linked to microbial community changes. We will moderate our previous statements in the updated manuscript.

Specific comments*: There seems to be a lack of consistent acronym for aerobic methane oxidation in our community. I feel that the acronym chosen here (MOx) is not specific enough to aerobic methane oxidation. Perhaps AMOx could be used instead?*

Reply: "MOx" as an abbreviation for aerobic oxidation of methane is quite widely used (and distinct from the acronym AOM, which is used for the anaerobic modes of methane oxidation); see for example Mau et al. 2016, BG; James et al. 2016, L&O; Pack et al. 2015 JGR Biogeosciences; Steinle et al. 2015, Nature Geoscience; Lofton et al. 2014, Hydrobiologia, Niemann et al. 2006, Nature. For consistency we prefer to keep MOx as an abbreviation. Its first use in the text is be explained.

*Fig.1 I'm not sure if it's just my pdf version but the figure is incredibly small and it could be more detailed*

Reply: We will improve the figure accordingly.

*Page 4 line 16: change determinations to measurements*

Reply: We will change that.

*Page 4 line 26: what is the in situ temperature? Did it change seasonally?*

Reply: Yes, the in situ temperature changed seasonally, and with depth (Figure 2). Samples

were incubated at the corresponding in situ temperature (different for instance, below and above the thermocline). Sample locations and corresponding in-situ temperatures are already depicted in Figure 2. We will refer to Figure 2 on line 26 of Page 4.

*Page 6 line 3: what temperatures exactly?*

Reply: The different temperatures are provided Figures 5 and S2. This will be clarified in the revised text.

*Page 10 line 32: insert "in the Baltic Sea" between "evidence that MOB" and "are well adapted".*

Reply: We will do that.

*Figure caption for Fig 3: panel (d) is mislabelled (b). Alternatively to 3c and 3d, a table with initial and final O2 concentrations for both sets of experiments might be more informative?*

Reply: We will correct this in the revised version. We will add a table with MOx rates, O2 consumption rates etc. in the supplementary section.

[revised manuscript text omitted]

Lea Steinle 2.2.2017 15:42

Lea Steinle 2.2.2017 15:42

Lea Steinle 2.2.2017 15:43

Lea Steinle 2.2.2017 15:51
**Moved (insertion) [1]**

Lea Steinle 24.2.2017 17:54

Lea Steinle 24.2.2017 17:55

Lea Steinle 24.2.2017 17:54

Lea Steinle 2.2.2017 15:53

Lea Steinle 2.2.2017 15:44

Lea Steinle 7.2.2017 11:37

Lea Steinle 7.2.2017 11:37

Lea Steinle 7.2.2017 11:37

Lea Steinle 7.2.2017 11:29

Lea Steinle 2.2.2017 15:51
**Moved up [1]:** In the subsequent discussion of the experimental data, for the sake of simplicity, hypoxic conditions (i.e., $[O_2]$ <63 µmol $l^{-1}$) will be referred to as "low" and $[O_2]$ >63 µmol $l^{-1}$ as "high" oxygen concentrations. .

**3.4 Temperature dependence of MOx**

In general, $k$ increased with temperature and reached maximum values at 20–37 °C, indicating a mesophilic temperature optimum (Fig. 5a,c; shown are results from Sept. 2013 and Feb. 2014; Table 1; Fig. S3). Only in Nov. 2013, maximum MOx was observed at 10–20 °C, consistent with a psychrophilic temperature optimum (Fig. 5b). These patterns were independent of water depth (Fig. 5, Fig. S3).

**3.5 Water-column methane removal by MOx and methane fluxes to the atmosphere**

Depth-integrated $r_{MOx}$ (= $F_{MOx}$) varied between 11.7 µmol m$^{-2}$ d$^{-1}$ (Mar. 2013) and 82.3 µmol m$^{-2}$ d$^{-1}$ (Sept. 2014; Table 2). Estimated average fluxes of methane to the atmosphere were 3.6–9.2 µmol m$^{-2}$ d$^{-1}$ during stratified periods, and 10.0–25.1 µmol m$^{-2}$ d$^{-1}$ during mixed periods (considering a minimum or maximum $k_w$, respectively; Table 2; Raymond and Cole 2001). According to the buoyancy frequency $N$, we grouped the sampling dates into two categories: weakly (i.e., N <120 cph) and strongly stratified (i.e., N >120 cph). The water column was only weakly stratified in Oct. 2012, Mar. 2013, and Feb. 2014, and strongly stratified during all other samplings (Table 2).

**4 Discussion**

**4.1 Seasonal variations at Boknis Eck**

**4.1.1 Development of seasonal hypoxia**

Oxygen concentrations were always close to saturation levels during our winter samplings (i.e., Mar. 2013, Feb. 2014), when the water column was poorly stratified and phytoplankton blooms occurred, which is typical for this time period (Bange et al., 2010). During June samplings, we observed much lower bottom water oxygen concentrations, indicating the onset of hypoxia, reaching sub-micromolar oxygen concentrations below 24 mbsl in September (2013, 2014). Our observation is in accordance with a previous time series study (2006–2008) by Bange et al. (2010), who found hypoxic events starting between May and August and lasting until September or November. Long-term monitoring at Boknis Eck showed that the frequency and length of hypoxic events have increased over the last twenty years (Lennartz et al., 2014), although nutrient inputs into the Baltic Sea were strongly reduced (HELCOM, 2009). One of the main reasons for the on-going decrease in oxygen concentration is the increasing water temperature since the 1960s (Lennartz et al., 2014). Higher surface water temperatures have led to an extension of the stratification period (starting earlier in the annual cycle), reducing the overall exchange between bottom and surface waters (Hoppe et al., 2013; Lennartz et al., 2014). Furthermore, a general increase in temperature also enhances mineralisation of organic matter in bottom waters and results in a higher biological oxygen demand (Hoppe et al., 2013; Lennartz et al., 2014).

Lea Steinle 7.2.2017 11:37

Lea Steinle 7.2.2017 11:37

Lea Steinle 24.2.2017 17:39

Lea Steinle 24.2.2017 17:40

**4.1.2 Seasonal dynamics of methane concentrations and MOx**

Methane concentrations at Boknis Eck were in a similar range as in other shelf seas and coastal ecosystems (e.g., Rehder et al., 1998; Bange, 2006; Upstill–Goddard 2016). We did not observe any clear seasonal methane concentration patterns as observed for oxygen concentrations and other physico-chemical parameters (Fig. 2). Our data are consistent with

5 observations from 2006–2008 by Bange et al. (2010), who showed that methane concentrations did not follow bimodal seasonal variations. Instead, increases in water column methane followed chlorophyll *a* concentrations in surface waters with a 1-month time lag, suggesting that pulses of elevated organic matter input to the sediments were boosting benthic methanogenesis.

We did not observe direct links between water column oxygenation and methane concentrations ($R^2 = 0.007$; $p = 0.6$).

[revised manuscript text omitted]

Lea Steinle 14.2.2017 11:30

Lea Steinle 14.2.2017 20:53

**Formatted** … [6]

Lea Steinle 14.2.2017 20:53

**Formatted** … [8]

Lea Steinle 14.2.2017 11:34

Lea Steinle 14.2.2017 20:53

**Formatted** … [9]

Lea Steinle 14.2.2017 11:34

Lea Steinle 14.2.2017 20:53

**Formatted** … [10]

Lea Steinle 14.2.2017 11:34

Lea Steinle 14.2.2017 20:53

**Formatted** … [12]

Lea Steinle 14.2.2017 11:34

Lea Steinle 14.2.2017 20:53

**Formatted** … [14]

Lea Steinle 25.2.2017 08:55

Lea Steinle 14.2.2017 20:53

**Formatted** … [15]

Lea Steinle 14.2.2017 11:37

Lea Steinle 14.2.2017 20:53

**Formatted** … [17]

Lea Steinle 25.2.2017 08:58

Lea Steinle 14.2.2017 20:53

**Formatted** … [18]

Lea Steinle 14.2.2017 11:38

Lea Steinle 14.2.2017 20:53

**Formatted** … [19]

Lea Steinle 14.2.2017 11:38

Lea Steinle 14.2.2017 20:53

**Formatted** … [21]

Lea Steinle 14.2.2017 11:42

Lea Steinle 14.2.2017 20:53

**Formatted** … [22]

Lea Steinle 14.2.2017 17:49

Lea Steinle 14.2.2017 20:53

**Formatted** … [23]

Lea Steinle 14.2.2017 17:49

Lea Steinle 14.2.2017 20:53

**Formatted** … [24]

Lea Steinle 14.2.2017 17:49

Lea Steinle 14.2.2017 17:50

[revised manuscript text omitted]